



# Regional data sets of high-resolution (1 and 6 km) irrigation estimates from space

Jacopo Dari[1, 2, *], Luca Brocca[2], Sara Modanesi[2], Christian Massari[2], Angelica Tarpanelli[2], Silvia Barbetta[2], Raphael Quast[3], Mariette Vreugdenhil[3], Vahid Freeman[4], Anaïs Barella-Ortiz[5], Pere Quintana-Seguí[5], David Bretreger[6], Espen Volden[7]

[1] Dept. of Civil and Environmental Engineering, University of Perugia, Perugia, Italy
[2] Research Institute for Geo-Hydrological Protection, National Research Council, Perugia, Italy
[3] Department of Geodesy and Geoinformation, Research Unit Remote Sensing, TU Wien, Vienna, Austria
[4] Earth Intelligence, Spire Global, 2763 Luxembourg, Luxembourg
[5] Observatori de l'Ebre (OE), Ramon Llull University - CSIC, 43520 Roquetes, Spain
[6] School of Engineering, The University of Newcastle, Callaghan, New South Wales 2308, Australia
[7] European Space Agency, ESRIN, Frascati, Italy

*Correspondence to*: Jacopo Dari (jacopo.dari@unipg.it)

**Abstract.** Irrigation water use represents the primary source of freshwater consumption by humans. The amount of water withdrawals for agricultural purposes is expected to further increase in the upcoming years to face the rising world population and higher living standards. Hence, effective plans for enacting a rational management of agricultural water use are urgent, but they are limited by knowledge gaps about irrigation. Detailed information on irrigation dynamics (i.e., extents, timing, and amounts) is generally lacking worldwide, but satellite observations can be used to fill this gap.

This paper describes the first regional-scale and high-resolution (1 and 6 km) irrigation water data sets obtained from satellite observations. The products are developed over three major river basins characterized by varying irrigation extents and methodologies, as well as by different climatic conditions. The data sets are an outcome of the European Space Agency (ESA) Irrigation+ project. The irrigation amounts have been estimated through the SM-based (Soil-Moisture-based) inversion approach over the Ebro river basin (North-eastern Spain), the Po valley (Northern Italy), and the Murray-Darling basin (South-eastern Australia). The satellite-derived irrigation products referring to the case studies in Europe have a spatial resolution of 1 km, and they are retrieved by exploiting Sentinel-1 soil moisture data obtained through the RT1 (first-order Radiative Transfer) model. A spatial sampling of 6 km is instead used for the Australian pilot area, since in this case the soil moisture information comes from CYGNSS (Cyclone Global Navigation Satellite System) observations. All the irrigation products are delivered with a weekly temporal aggregation. The 1 km data sets over the two European regions cover a period ranging from January 2016 to July 2020, while the irrigation estimates over the Murray-Darling basin are available for the time span April 2017 – July 2020. The retrieved irrigation amounts have been compared with benchmark rates collected over selected agricultural districts. Results highlight satisfactory performances over the major part of the pilot sites falling within the two regions characterized by a semi-arid climate, namely the Ebro and the Murray-Darling basins, quantified by median values of $RMSE$, Pearson correlation, $r$, and $BIAS$ equal to 12.4 mm/14-day, 0.66, and -4.62 mm/14-day, respectively, for



the Ebro basin and to 10.54 mm/month, 0.77, and -3.07 mm/month, respectively, for the Murray-Darling basin. The
assessment of the performances over the Po valley is affected by the limited availability of in situ reference data for
irrigation.

## 1 Introduction

Human activities are deeply modifying the natural hydrological cycle, as they affect water storage and relocation dynamics.
Among the anthropogenic water uses, the agricultural one (mainly constituted of water applied for irrigation) prevails. More
than 70% of surface and sub-surface water withdrawals worldwide are destined to irrigation practices (Foley et al., 2011;
Dorigo et al., 2021). In the near future, the pressure exerted on water resources to foster irrigation is expected to be further
exacerbated to face the global challenge represented by the increasing food demand due to the population growth under
climate warming scenarios (Hunter et al., 2017; Ferguson et al., 2018). Hence, strategies aimed at a rational management of
agricultural water are essential for several environmental, economic, and social reasons, as well as to comprehensively
characterize the hydrological cycle over anthropized basins. Nevertheless, irrigation practices are scarcely monitored
worldwide, thus creating the paradoxical situation for which the largest human intervention on the water cycle is mostly
unknown. Remote sensing technology offers unprecedented opportunities for answering the following research question:
how much water is used for irrigation? More in details, remotely sensed observations of soil moisture and evapotranspiration
are particularly suitable for the development of irrigation quantification techniques, as demonstrated by a number of recent
studies implementing evapotranspiration-based (see, e.g., Romaguera et al., 2014; Van Eekelen et al., 2015; Peña-Arancibia
et al., 2016; Brombacher et al., 2022) and soil-moisture-based (see, e.g., Brocca et al., 2018; Jalilvand et al., 2019;
Zaussinger et al., 2019; Dari et al., 2020, 2022b; Zappa et al., 2021; Zhang et al., 2022) approaches. Brocca et al. (2018) first
proposed an irrigation quantification methodology relying on the inversion of the satellite soil moisture signal, currently
known as the SM-based (Soil-Moisture-based) inversion approach. The feasibility of estimating irrigation rates through
coarse resolution soil moisture data from AMSR2 (Advanced Microwave Scanning Radiometer 2), ASCAT (Advanced
SCATterometer), SMAP (Soil Moisture Active Passive), and SMOS (Soil Moisture and Ocean Salinity) was demonstrated,
even though limitations due to the low spatial resolution of the input data sets were pointed out. Later on, the methodology
was implemented by Jalilvand et al. (2019) over a semi-arid region of Iran by exploiting AMSR2, ASCAT, and SMOS soil
moisture data. In this study, the not-negligible contribution of evapotranspiration in properly reproducing irrigation amounts
was highlighted. This result was corroborated by Dari et al. (2020), who further developed the method by implementing the
guidelines provided by the FAO (Food and Agriculture Organization) paper n.56 (Allen et al., 1998) within the SM-based
inversion algorithm to develop a finer modeling of crop evapotranspiration. The approach was successfully applied over
heavily irrigated areas located in North-eastern Spain. In that case, input soil moisture data from the 1 km resolution
DISPATCH (DISaggregation based on Physical And Theoretical scale CHange; Merlin et al., 2013) downscaled SMOS and
SMAP products were used. Dari et al. (2022b) further studied the role of the evapotranspiration component within the SM-



based inversion algorithm and introduced the interesting perspective of a configuration forced with remotely sensed data only. The SM-based inversion approach is not the unique method relying on soil moisture data to estimate irrigation existing in literature. Another example is represented by the SM-delta (Soil-Moisture-delta) algorithm, first proposed by Zaussinger et al. (2019). The approach is based on the discrepancies between satellite and modeled soil moisture, which, over agricultural areas, can be attributed to irrigation signals. The method was later updated by Zappa et al. (2021), who added terms accounting for evapotranspiration and drainage fluxes to the algorithm. This new configuration was applied over agricultural fields in Northern Germany; high-resolution (500 m) soil moisture observations from Sentinel-1 were used and an agreement between estimated irrigation amounts and benchmark rates quantified by an average Pearson correlation equal to 0.64 was found. The SM-delta was recently applied in synthetic and real-world experiments to highlight the mutual effects of varying satellite soil moisture temporal and spatial resolutions on the accuracy of the retrieved irrigation amounts (Zappa et al., 2022). Finally, data assimilation (DA) techniques exploiting remote sensing soil moisture can also be used in irrigation quantification studies to balance modeling deficiencies and correct unrealistic assumptions. Indeed, models have recently seen improvements in irrigation parameterization (Ozdogan etal., 2010; Lawston et al., 2015; Nie et al., 2018). However, some studies (e.g., Modanesi et al., 2021b) have highlighted that the performance of LSM (Land Surface Model) irrigation simulations are negatively affected by simplified assumptions in model parameterizations, and by unrealistic/out of date input information (e.g., lack of dynamic crop maps). DA can reduce models' uncertainties by merging model systems with satellite observations, which can track human-induced processes. In particular, Lawston et al. (2017) suggested the use of SMAP surface soil moisture retrievals to incorporate the irrigation signal into models via DA and more recently Jalivand et al. (2021) exploited the potential of SMAP/Sentinel-1 retrieval (Das et al., 2019) for the same purpose. In this context, Abolafia-Rosenzweig et al. (2019) designed a DA system for remote-sensing-based soil moisture assimilation into the VIC (Variable Infiltration Capacity) model (Liang et al., 1994) to improve irrigation estimates through a particle batch smoother. An alternative way to assimilate satellite observations is to directly ingest level-1 observations (i.e., brightness temperature or radar backscatter), instead of retrievals (De Lannoy and Reichle, 2016a, 2016b; Lievens et al., 2017a, 2017b; Modanesi et al. 2022). In particular, Modanesi et al. (2022) assimilated 1 km Sentinel-1 backscatter ($\gamma^0$) observations into the Noah MP LSM, equipped with a sprinkler irrigation scheme into the National Aeronautics and Space Administration (NASA) Land Information System (LIS) framework, for the update of both surface soil moisture and vegetation states. The authors found that DA improves the bias of irrigation simulation although limitations mainly due to irrigation model parameterization still need further improvement.

The high-resolution retrievals of the latest satellite capabilities open unprecedented perspectives in the irrigation quantification activity. As pointed out in Peng et al. (2021), the optimal spatial resolution for monitoring agricultural practices is less than or equal to 1 km. This is particularly true in the Mediterranean area, where the nominal size of the agricultural fields makes the adoption of high-resolution data necessary (Dari et al., 2022a).

This study is aimed at presenting three regional-scale, high-resolution irrigation data sets developed through the SM-based inversion approach within the European Space Agency (ESA) Irrigation+ project (https://esairrigationplus.org/). The



irrigation products have been developed for two European regions, namely the Ebro basin (Spain) and the Po valley (Italy), and for the Murray-Darling basin (Australia). Both data sets referring to the Mediterranean area rely on input soil moisture from the RT1 (first-order Radiative Transfer) Sentinel-1 data set (Quast et al., 2019), are delivered over a 1 km regular grid, and cover the period ranging from January 2016 to the end of July 2020. For the Australian sites, Spire's soil moisture product retrieved from NASA's CYGNSS (Cyclone Global Navigation Satellite System) satellites data (Freeman et al., 2020) has been used instead; irrigation estimates are sampled over a 6 km regular grid and the temporal coverage is from April 2017 to July 2020. All the products are delivered with a weekly temporal aggregation. Even though an assessment of the product's performances is provided by exploiting benchmark irrigation amounts over selected sites, the authors' main goal is to make the data sets publicly available as they can be used and further validated by the scientific community. The proposed irrigation estimates represent an important step towards the implementation of an operational system for high-resolution irrigation water monitoring from satellite observations.

## 2 Pilot areas

The irrigation products have been developed over three regions highly influenced by irrigation practices. They are the Ebro river basin (~86000 km$^2$) in Spain, the Po valley (~78000 km$^2$) in Italy, which includes the Po river basin and part of the Emilia-Romagna region, and the Murray-Darling basin (~1000000 km$^2$) in Australia. According to the Köppen-Geiger climate classification (Beck et al., 2018) provided in Figure 1, the Ebro basin is mainly characterized by a cold semi-arid climate (BSk), with oceanic climate areas (Cfb) interesting the upper part. The Po river valley mainly falls within the humid subtropical climatic zone (Cfa), while the Murray-Darling basin is mainly subject to hot and cold semi-arid climatic areas (BSh and BSk, respectively); desert climate (BWh) is also present for a minor extent.

The Ebro basin is the largest Mediterranean basin of Spain and a major Mediterranean basin in Europe. Precipitation is unevenly distributed, being higher in the mountainous regions, where it can reach 1800 mm/year, and lower in the central valley, with values below 500 mm/year. Therefore, to irrigate agricultural areas, mainly located in the central valley, there is an extensive network of dams and canals to transport water from the mountains to these regions. For instance, the total dam capacity is approximately 8000 hm$^3$ (PHE 2015-2021[1]). Focusing on agriculture, the most representative herbaceous crops are alfalfa, corn, barley, wheat, and rice, while the most representative tree crops are peach and pear trees, vineyards, and olive groves. There are almost 9660 km$^2$ conceded for irrigated surface over the basin, being the average agricultural annual demand under objective conditions of 7623 hm$^3$/year.

The Po valley is part of the Po river basin, the longest river in Italy. The key importance in terms of agricultural production together with the high sensitivity to recent severe drought events (Strosser et al., 2012; Ceppi et al., 2014; Formetta et al., 2022) have turned the Po valley into a critical hotspot for studying the water assessment and impact of human activities on

---

[1] Ministerio de Agricultura, Alimentación y Medio Ambiente, Confederación Hidrográfica del Ebro: Plan Hidrológico de la parte española de la demarcación hidrográfica del Ebro 2015-2021. v2.6 Memoria, 2015



130  the water cycle. The Northern side of the valley has a higher water availability compared to the South, thanks to the presence of several Alpine reservoirs (Musolino et al., 2017). In the South, the Emilia Romagna region is poorer in storage capacity but it is served by one of the most important Italian hydraulic systems for irrigation applications, the *Canale Emiliano Romagnolo* (CER, https://consorziocer.it/it/). The size and spatial extent of the irrigated fields and districts in the Po valley are often not homogeneous and agricultural plots are characterized by small extents due to complex historical processes

135  (Massari et al., 2021). The main cultivated crops include general summer and winter crops, orchards, olive groves, and vineyards (https://sites.google.com/arpae.it/servizio-climatico-icolt/home?authuser=0, last access: 4 November 2022).

The Murray-Darling basin is often considered the food bowl of Australia, covering over 1000000 km$^2$ or approximately 14% of Australia and accounts for over two thirds of all of Australia's irrigation water use. There has been a cap on diversions since 1995 to help manage over allocation and extraction in the basin. The basin is often subject to extreme droughts such as

140  the Millennium drought (2001-2009) (van Dijk et al., 2013) and more recently the 2017-2019 drought. There is typically more irrigation in the Southern side of the basin which is facilitated by major storages in the region. Irrigated properties are often fragmented in nature and contain a wide range of crops that are suited to the vastly different conditions that are observed across such a large basin. Both surface water and groundwater are used across the basin with groundwater use mainly associated with major alluvial systems.

145

**Figure 1: Location of the pilot areas and their climatic characteristics according to the Köppen-Geiger classification. The Ebro basin and the Po valley are indicated by the number 1 and 2, respectively. The Murray-Darling basin is identified by the number 3.**

## 3 Materials and methods

### 3.1 The SM-based inversion approach

Regional-scale irrigation products have been developed through the SM-based inversion approach (Brocca et al., 2018; Dari et al., 2020; 2022b). The method relies on the inversion of the soil moisture signal for backward estimating the total amount



of water entering into the soil, which, over agricultural areas, is determined by rainfall plus irrigation. The approach is based
on the soil water balance, expressed by:

$$Z^* \frac{dS(t)}{dt} = i(t) + r(t) - g(t) - sr(t) - e(t) \tag{1}$$

where $Z^*$ [mm] indicates the water capacity of the soil layer, calculated as the product between the depth of the soil layer and
the porosity, $S(t)$ [-] is the relative soil moisture (i.e., ranging between 0 and 1), $t$ [days] indicates the time, $i(t)$ is the
irrigation rate [mm/day], $r(t)$ [mm/day] is the rainfall rate, $g(t)$ [mm/day] indicates the drainage term, $sr(t)$ [mm/day] is
the surface runoff, and $e(t)$ [mm/day] represents the evapotranspiration rate. The Eq. (1) is equivalent to the following:

$$Win(t) = Z^* \frac{dS(t)}{dt} + g(t) + sr(t) + e(t) \tag{2}$$

with $Win(t)$ indicating the algorithm output, i.e. the total amount of water entering into the soil. The drainage term can be
linked to soil moisture according to the power law equation $g(t) = aS(t)^b$, in which $a$ and $b$ are drainage parameters
(Famiglietti and Woods, 1994; Brocca et al., 2014). As demonstrated in previous studies, the $sr(t)$ term can be neglected
(Brocca et al., 2015; Jalilvand et al., 2019), since irrigation water either infiltrates or evaporates. The actual
evapotranspiration contribution is computed as the potential rate, $PET(t)$, limited by the available water content: $e(t) = F \cdot S(t) \cdot PET(t)$; an adjustment factor, $F$, ranging between 0.6 and 1.4 and aimed at accounting for uncertainties linked to the
coarse resolution of the input PET rates is adopted (Modanesi et al., 2021a). Hence, Eq. (2) can be simplified as:

$$Win\ (t) = Z^* \frac{dS(t)}{dt} + aS(t)^b + F \cdot S(t) \cdot PET(t) \tag{3}$$

Before running the algorithm, the noise in the soil moisture signal is reduced by computing the Soil Water Index (SWI)
according to the exponential filter proposed by Albergel et al. (2008). Once the total amount of water entering into the soil is
quantified, it is possible to obtain the irrigation rate by removing the rainfall from the output of Eq. (3), $i(t) = Win\ (t) - r(t)$; negative irrigation rates (if any) are set equal to zero. In order to remove negligible irrigation amounts attributable to
random errors, the results are discarded if the ratio between weekly estimated irrigation and weekly rainfall is lower than 0.2.
Weekly irrigation estimates covering the time span from January 2016 to July 2020 have been produced for the Ebro basin
and the Po river valley. Over the Murray-Darling basin, weekly irrigation amounts have been retrieved by considering the
period from April 2017 to July 2020. A regular 1 km grid has been adopted for the Spanish and the Italian case study, while
a 6 km sampling has been used for the Australian pilot basin. For each region, the irrigation amounts have been estimated
over agricultural areas only. For the Ebro and Po regions, such an information has been derived by the 25 m resolution
Corine Land Cover data set referring to the year 2018 (CLC2018), while for the Murray-Darling basin the 300 m spatial
resolution ESA CCI (Climate Change Initiative) land cover map for the year 2018 has been exploited.

The parameters $a$, $b$, $Z^*$, and $F$ of Eq. (3) are calibrated by implementing the iterative procedure summarised in Figure 2.
First, the algorithm is run by masking out days with no rainfall rate during the irrigation seasons (hence, potential irrigation
days). This first step involves the $a$, $b$, and $Z^*$ parameters which are optimized by minimizing the Root Mean Square
Difference ($RMSD$) against reference rainfall rates; during this phase, implemented for each pixel, the evapotranspiration



adjustment factor, $F$, is assumed equal to 1. Then, $F$ is calibrated against the sum of rainfall plus irrigation over selected pilot sites where irrigation rates are known. At this point, the calibrated $F$ value is compared with the first guess ($F = 1$) and, in case of disagreement, the values of $a$, $b$, and $Z^*$ and are re-calibrated by repeating the first step and adopting the updated $F$ value. It is important to highlight that the calibration of $a$, $b$, and $Z^*$ is spatially distributed, hence, such values differ pixel

by pixel. Conversely, the calibration of $F$ can be only implemented over selected sites where irrigation rates are known and for this reason spatially aggregated time series must be used. The period 2016-2017 has been considered for the calibration procedure over the Ebro basin and the Po valley, while for the Murray-Darling catchment the time span ranging from April 2017 to the end of 2018 has been adopted. The $F$ parameter has been calibrated over three districts in the Ebro basin, two small districts in the Po river valley and three districts in the Murray-Darling basin. For each region, the areal-weighted

average of the $F$ values has been adopted (as a fixed parameter for all the pixels). For more details on the outputs of the calibration procedure, as well as on the irrigation data used to calibrate the $F$ parameter, the reader is referred to Appendix A and Sub-section 3.3, respectively.

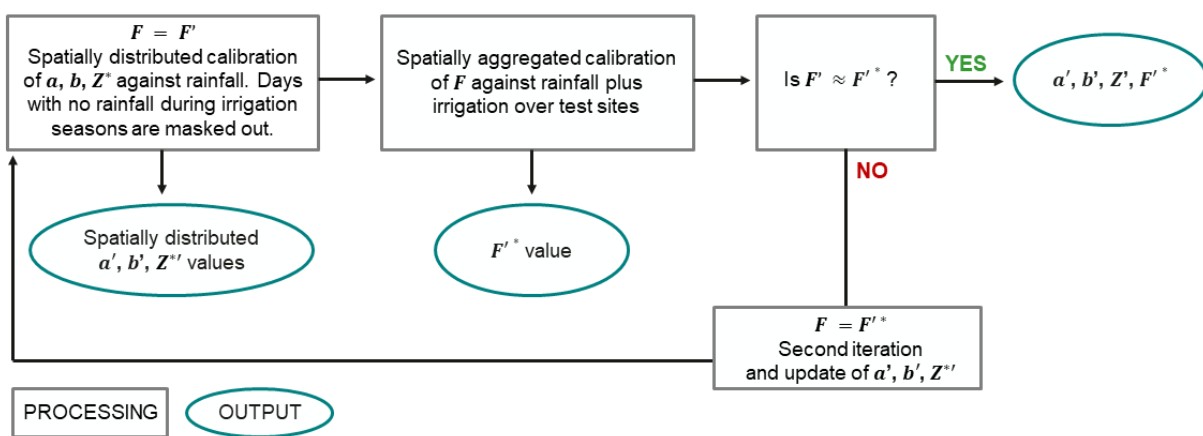

**Figure 2: Iterative procedure adopted to calibrate the SM-based inversion algorithm parameters $a, b, Z^*$, and $F$.**

**3.2 Input data sets**

Input time series of soil moisture, potential evapotranspiration (PET), and rainfall are needed to run the SM-based inversion approach. For the Ebro and the Po regions, the irrigation products rely on 1 km soil moisture data derived by Sentinel-1 observations. More in detail, Soil moisture is obtained from incidence angle dependent Sentinel-1 backscatter measurements

at 500m spatial sampling (~1 km spatial resolution) (Bauer-Marschallinger et al., 2019) by using a time series based first-order radiative transfer modelling approach (RT1) (Quast et al., 2019; 2023). The scattering characteristics of soil- and vegetation are hereby modelled via parametric distribution functions. The retrieval is then performed via a non-linear least-squares regression procedure that minimises the difference between the measured and modelled backscatter time series for



each pixel individually (Quast, 2021). To account for spatial variations in the soil-scattering characteristics, a temporally
constant soil-scattering directionality parameter is optimised. To correct for effects induced by seasonal dynamics of
vegetation, auxiliary Leaf Area Index (LAI) time series provided by ECMWF (European Centre for Medium-Range Weather
Forecasts) ERA5-Land (European ReAnalysis v5-Land) reanalysis data set (Muñoz-Sabater et al., 2021) are used as proxy
for the seasonal dynamics of the vegetation optical depth. In addition, the single-scattering albedo of the vegetation layer is
optimised for each Sentinel-1 orbit individually to account for differences in the observation geometry between consecutive
timestamps. The soil moisture time series is finally obtained by implying a linear relationship to the normalisation factor of
the bare-soil scattering distribution function. The resulting soil moisture data set therefore represents a percentage measure
of the relative moisture saturation of the soil surface.

The irrigation product developed over the Murray-Darlig basin relies on CYGNSS observations. The GNSS-R remote
sensing is a relatively new technique based on a bistatic radar system that is used to perform Earth surface scatterometry.
The applications of GNSS-R technique have been demonstrated in several studies using the data from recent space-borne
missions, such as TechDemoSat-1, CYGNSS, and Spire's GNSS-R satellites[2]. The CYGNSS satellites constellation consists
of eight GNSS-Reflectometry (GNSS-R) satellites launched in December 2016 into a 35-degree inclination low Earth. Each
satellite carries a 4-channel GNSS-R bistatic radar receiver tuned to receive the L1 signals transmitted by GPS (Global
Positioning System) satellites (Ruf et al. 2017). The Spire soil moisture retrieval algorithm is based on a change detection
method (Freeman et al. 2020). GNSS-R reflectivity measurements over land vary depending on soil water content, surface
roughness, and vegetation. The vegetation and roughness changes occur on timescales longer than soil moisture changes
which makes it possible to monitor soil moisture in the presence of vegetation and surface roughness contributions. For a
given geographic location, the surface roughness can be viewed as almost constant. Seasonal changes in vegetation can still
affect reflection, but on a much longer time scale than changes in soil moisture. Short-term fluctuations in reflectivity (dB
scale) are roughly linearly related to changes in soil moisture. A relative measure of reflectivity that corresponds to
variations in soil moisture levels can be calculated by scaling the normalised reflectivity between the lowest and highest
reflectivity measurements (dry and wet references in each location) that correspond in each case on the vegetation wilting
point and the degree of soil saturation. The obtained relative soil moisture measurements were calibrated using the
concurrent SM measurements from NASA's SMAP mission. In this study, Spire's CYGNSS-based SM product retrieved
from GNSS-R observations on a 6 km equidistant grid in the Murray-Darling basin has been used.

Finally, PET rates from the GLEAM (Global Land Evaporation Amsterdam Model) v3.5b product at 0.25° spatial resolution
(Miralles et al., 2011; Martens et al., 2017) have been used to force the algorithm, while rainfall rates have been derived by
the ERA5-Land data set at 9 km spatial resolution.

---

[2] The Spire GNSS-R data are available through the NASA CSDAP and ESA Earthnet programs.
https://earth.esa.int/eogateway/missions/spire
https://www.earthdata.nasa.gov/esds/csda/commercial-datasets



All the data sets used have been resampled to the adopted regular grids (1 km for Ebro and Po bsins, 6 km for Murray-
Darling basin) as a pre-processing step.

## 3.3 Benchmark irrigation amounts

Benchmark irrigation rates over selected sites have been collected for calibration and validation purposes. Within the Ebro basin, four districts have been considered: Urgell (887.6 km$^2$), Algerri Balaguer (70.8 km$^2$), North Catalan and Aragonese (657.0 km$^2$) and South Catalan and Aragonese (504.5 km$^2$). The districts differ from each other in terms of irrigation

techniques and management (Dari et al., 2021). The dense network of irrigation canals feeding the districts is monitored by the SAIH (*Sistema Automático de Información Hidrológica,* http://www.saihebro.com/saihebro/index.php) system of the Ebro river basin, which provides data about water volumes flowing through the canals. Hence, this information has been collected at a daily temporal scale for the period 2016-2019. For each district, the irrigation doses in millimetres have been calculated by dividing the volumes by the areas of interest. Losses due to irrigation efficiency have been considered as

described in Dari et al. (2020).

Two small districts have been considered for the Po valley: San Silvestro (2.9 km$^2$) and Formellino (7.6 km$^2$). They are located around the city of Faenza, in the Emilia Romagna region. For each pilot district, daily irrigation amounts (in mm) have been provided by the CER consortium for the period 2016–2017. The crops growing on the Faenza small-districts are mainly pear and kiwi trees.

Data from five irrigation districts located in New South Wales, Australia have been collected for the Murray-Darling basin: Coleambally (977.0 km$^2$), Murrumbidgee (2789.3 km$^2$), Western Murray (49.1 km$^2$), Murray Mulwala (3092.6 km$^2$), and Murray Wakool (1455.2 km$^2$). Each district is managed by an Irrigation Infrastructure Operator (IIO), which is responsible for the production of annual reports on the irrigation water withdrawals (Bretreger et al., 2020). It is noteworthy that the Murray Mulwala and Murray Wakool districts belong to the same IIO. Monthly irrigation amounts referring to the period

ranging from April 2017 to April 2019 have been considered. As in the case of the Ebro basin, the irrigation amounts in millimetres have been obtained by dividing the volumes provided in the IIO reports by the area of interest. It is noteworthy that minor portions of the Western Murray and of the Murrumbidgee districts fall outside the agricultural domain derived from the ESA CCI (Climate Change Initiative) land cover. Hence, for the abovementioned districts, the irrigation rates have been computed by considering the portion of area overlapped with the mask of agricultural pixels.

Figure 3 provides an overview on the location of all pilot irrigation districts with respect to the agricultural portions (white areas) over which the irrigation estimates have been produced. The ground-truth information on the collected irrigation rates is summarised in Table 1.

A part of the collected irrigation amounts has been used to calibrate the *F* parameter (see Sub-section 3.1). More in detail, irrigation data referring to the Algerri Balaguer and to the North and South Catalan Aragonese districts for the 2-year period

2016-2017 have been used to calibrate *F* over the Ebro basin. For the Po valley, given the limited information available, all the collected data have been used (i.e., irrigation applications over the San Silvestro and Formellino districts during the time





span 2016-2017). Finally, irrigation volumes for the period ranging from April 2017 to December 2018 and referring to the three districts, i.e., Coleambally, Murray Mulwala, and Murray Wakool, have been used for the Murray-Darling basin.

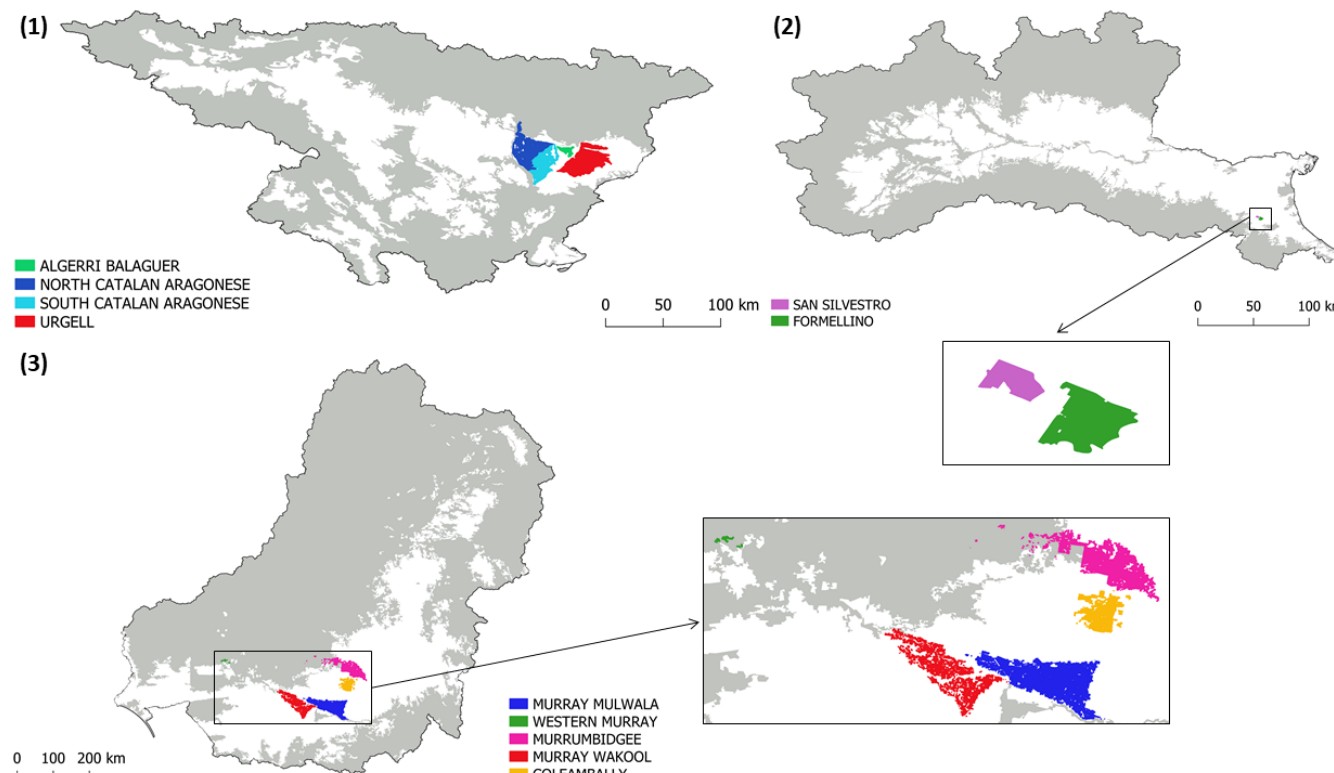


**Figure 3: Pilot irrigation districts whose irrigation data have been used for calibration and validation purposes: (1) the Ebro basin, (2) the Po valley, and (3) the Murray-Darling basin. For each region, the agricultural areas over which the irrigation estimates have been carried out are indicated in white.**

**Table 1: Summary of the collected ground-truth irrigation rates.**

| District | Area [$km^2$] | Time span of collected irrigation rates | Source |
|---|---|---|---|
| *Ebro basin* | | | |
| Urgell | 887.6 | | |
| Algerri Balaguer | 70.8 | | |
| North Catalan Aragonese | 657.0 | 2016-2019 | SAIH Ebro |
| South Catalan Aragonese | 504.5 | | |
| *Po valley* | | | |
| San Silvestro | 2.9 | | |
| Formellino | 7.6 | 2016-2017 | CER |





| *Murray-Darling basin* | | | |
|---|---|---|---|
| Coleambally | 977.0 | | |
| Murrumbidgee | 2789.3 | | |
| Western Murray | 49.1 | 2017 (April)-2019 (April) | IIOs |
| Murray Mulwala | 3092.6 | | |
| Murray Wakool | 1455.2 | | |

## 4 Results and discussion

In this Section, the retrieved irrigation amounts over each pilot area are presented and discussed. Comparisons with reference irrigation rates are provided as well. Sub-section 4.1 describes the irrigation product developed over the Ebro basin, Sub-section 4.2 is dedicated to the Po valley and Sub-section 4.3 refers to the Murray-Darling basin. The retrieved irrigation estimates are compared against the benchmark amounts in terms of $RMSE$ (Root Mean Square Error), Pearson correlation, $r$, and $BIAS$. Limitations of the proposed data sets and future plans are discussed in Sub-section 4.4 and 4.5, respectively.

### 4.1 Ebro basin, Spain

The data set of irrigation amounts retrieved through the SM-based inversion approach over the Ebro basin is developed over a 1 km regular grid and covers the period ranging from January 2016 to the end of July 2020. Figure 4 provides the maps of the cumulated irrigation estimates during the highest-intensity (May-September) irrigation seasons of 2016, 2017, 2018 and 2019. Some patterns of high irrigation rates recurring over areas known to be irrigated can be observed, as the pilot districts considered in this study (see Figure 3), the narrow portion unfolding from West to East along the main reach of the Ebro in the middle of the basin and the area close to the river delta.

**Figure 4: Cumulated irrigation amounts over the Ebro basin (Spain) during the highest-intensity irrigation season (May-September) of 2016, 2017, 2018, and 2019.**

Figure 5 (panels a-d) shows the 14-day aggregated time series of irrigation estimates (the black lines) over the pilot districts against the benchmark rates (the light grey shaded areas) for the period period 2016-2019; a yearly comparison in which data referring to different districts and years is indicated by different markers and colours, respectively, is provided as well (panel e). It is noteworthy that 2016 is not considered for the Urgell district, as there is a lack of irrigation benchmarks for half of the year. The time series in panels a)-d) highlight that the best performances in terms of *RMSE* are obtained over the North Catalan and Aragonese district (*RMSE* = 10.08 mm/14-day), while the best results in terms of *r* and *BIAS* are found for the Algerri Balaguer district ( *r* = 0.78 and *BIAS* = -2.23 mm/14-day). Even though a general tendency in slightly underestimating the benchmark amounts can be observed, the performances in terms of *RMSE* and *BIAS* are satisfactory, as the maximum deviation from previously mentioned minimum values is +4.58 mm/14-day for the *RMSE* and -6.59 mm/14-day for the *BIAS*; in both cases, the highest values of these two metrics refer to the Urgell district. Along with the irrigation





amounts, timing is also satisfactorily reproduced over the Algerri Balaguer ($r = 0.78$) and the South Catalan and Aragonese ($r = 0.74$) districts, while worst performances are observed over the remaining two districts. The results agree with previous

experiments carried out over a portion of the Ebro basin enclosing the same pilot districts of this study and in which irrigation estimates were retrieved by using a combination of DISPATCH SMOS and SMAP soil moisture with evapotranspiration rates calculated through the FAO56 approach (Dari et al., 2020). Moreover, the current implementation of a soil-moisture-limited approach to compute the actual evapotranspiration brings benefits in reproducing actual irrigation amounts with respect to previous attempts in this regard (Dari et al., 2022b).

Panel e) of Figure 5 provides a long-term comparison, in which the results are yearly cumulated; each point indicates the performance of a district in a certain year. A confidence interval of ± 30% of the benchmark is shown as well; according to Massari et al. (2021), such a value matches with the upper limit of the accuracy of satellite-derived irrigation products desired by farmers. In nine out of the fifteen total cases, the relative error is lower than ± 30%. Seasonal values referring to the Algerri Balaguer and the North Catalan and Aragonese districts are outside the interval only for one out of the four

considered years; underestimates lower than 30% of the total are found over the Urgell and the South Catalan and Aragonese districts for the years 2017 and 2019.





**Figure 5: Panel a)-d): 14-day aggregated time series of irrigation estimates (the black lines) over the four considered pilot districts**
**within the Ebro basin against the benchmark rates (the light grey shaded areas). Panel e): yearly comparison with data referring to different districts and years indicated through different markers and colours, respectively. The interval indicating a relative error of ± 30% is shown as well.**

## 4.2 Po valley, Italy

Similarly to the Ebro case study, the satellite-derived irrigation data set over the Po valley is developed over a 1 km regular
grid for the period ranging from January 2016 to the end of July 2020. The spatial distribution of the cumulated irrigation amounts during the highest-intensity irrigation seasons of 2016, 2017, 2018 and 2019 are provided in Figure 6. As a difference with the Ebro case study, strong contrasts between areas with very low and very high irrigation rates are not detected here. The only exception is a portion with low irrigation amounts in the South-eastern side that recurs in 2016, 2017 and 2019. This is an expected result, since while in the Ebro basin permanent rainfed and irrigated lands coexist, the Po
valley landscape consists of almost evenly distributed agricultural fields. Nevertheless, crop rotation creates a complex mosaic of irrigated and non-irrigated fields at a spatial scale that can be lower than 1 km.

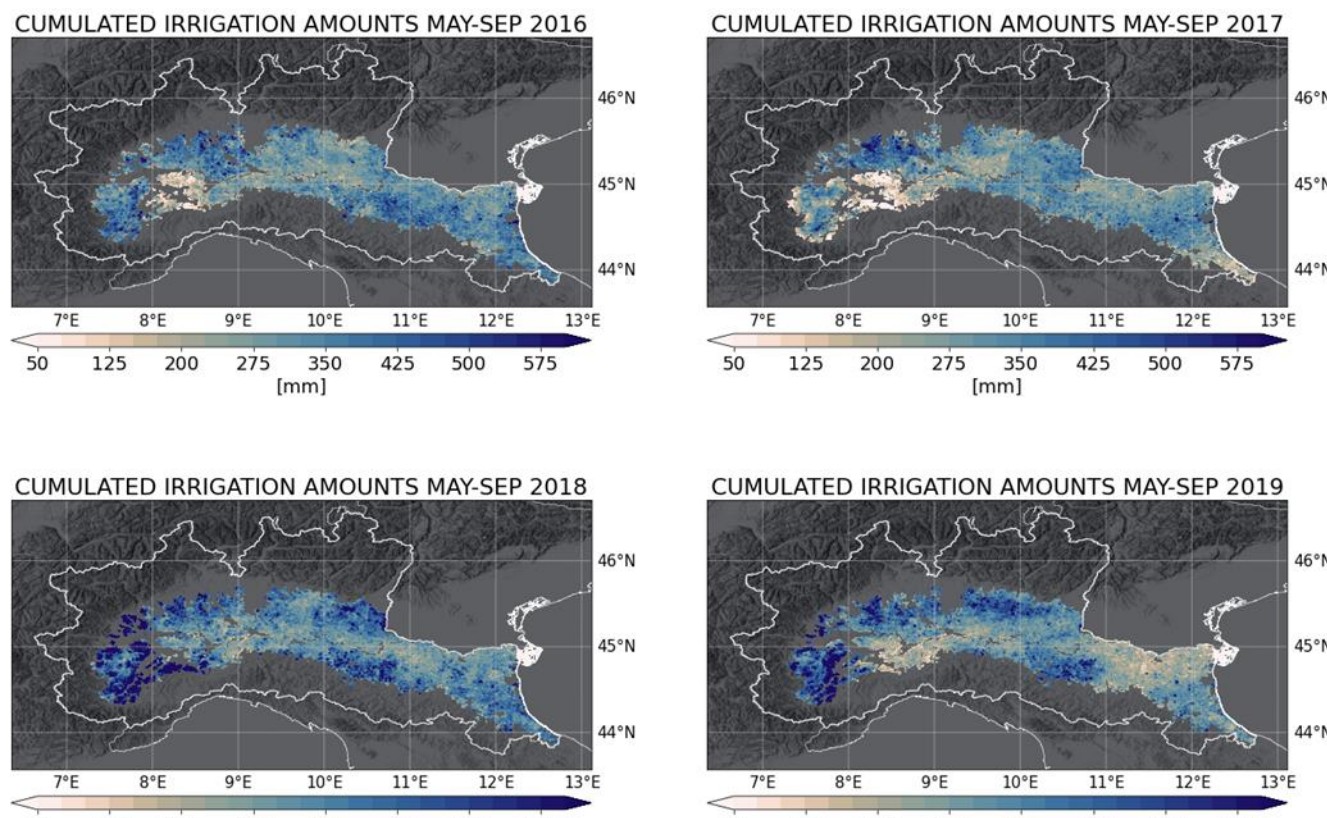

**Figure 6: Cumulated irrigation amounts over the Po valley (Italy) during the highest-intensity irrigation season (May-September)**
**of 2016, 2017, 2018, and 2019.**

The comparison between the proposed irrigation estimates and the benchmark rates collected over the San Silvestro and Formellino districts are provided in Figure 7. Panels a) and b) show the 14-day aggregated time series, while the yearly comparison is proposed in panel c). It is noteworthy that, as a difference with the Ebro case study, here the results during the non-irrigated season have been masked out. In fact, as shown by the time series of benchmark volumes of Figure 5, in the

Ebro basin irrigation can occur in winter as well, even though the volumes are much lower with respect to summer. It is the case, for instance, of irrigated fruit trees and greenhouses. For the pilot districts in the Po valley, the information on the irrigation season is available and it has been exploited. Hence, the metrics are computed during irrigation periods only. A slight tendency in overestimating the benchmark can be observed, the performances over Formellino ($RMSE$ = 10.90 mm/14-day and $BIAS$ = 3.54 mm/14-day) are more satisfactory with respect to San Silvestro ($RMSE$ = 17.77 mm/14-day

and $BIAS$ = 11.38 mm/14-day). In both cases, poor performances in reproducing irrigation timing are found ($r$ = 0.36 and $r$ = 0.32 for San Silvestro and Formellino, respectively). Panel c) of Figure 7 shows that, even though for both districts the benchmark yearly amounts are overestimated, the relative error for the representative points of Formellino is below + 30%. It is important to highlight that the validation over the Po valley is limited by the number and the size of the pilot districts. In





fact, it has been possible to collect benchmark irrigation rates over just two sites, each one enclosing a very limited number
of 1 km pixels of the satellite-derived irrigation data set. As a result, contamination with adjacent areas that can be non-
irrigated or irrigated but with mixed techniques can affect the results. Moreover, the 1 km resolution represents the upper
limit of the nominal size of agricultural fields over the Po valley; hence, the comparisons could suffer from a variability of
the irrigation dynamics occurring at the sub-pixel scale as well.

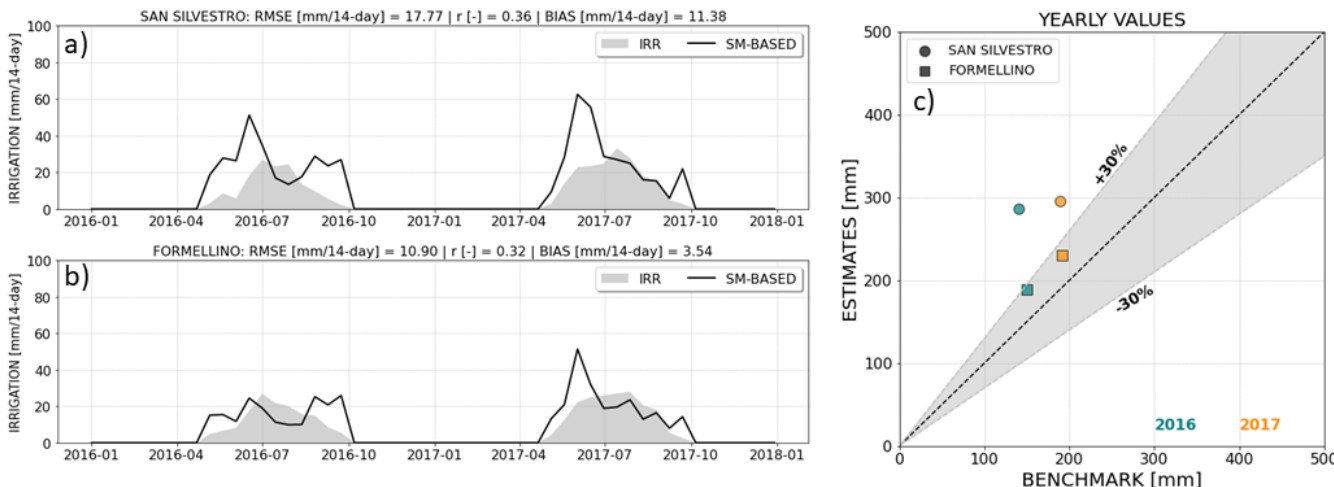

**Figure 7: Panel a)-b): 14-day aggregated time series of irrigation estimates (the black lines) over the two considered pilot districts within the Po valley against the benchmark rates (the light grey shaded areas). Panel c): yearly comparison with data referring to different districts and years indicated through different markers and colours, respectively. The interval indicating a relative error of ± 30% is shown as well.**

**4.3 Murray-Darling basin, Australia**

The irrigation data set over the Murray-Darling basin has been produced over a 6 km regular grid for the period ranging from
April 2017 to July 2020. The spatial distribution of the retrieved irrigation amounts cumulated during the highest-intensity
irrigation seasons (September-March) of 2017/18, 2018/19, and 2019/20 are shown in Figure 8. Recurring patterns of high
irrigation rates on the East and the South sides of the domain can be observed.




**Figure 8: Cumulated irrigation amounts over the Murray-Darling basin (Australia) during the highest-intensity irrigation season (September-March) of 2017/18, 2018/19, 2019/20.**

Figure 9 shows the comparisons between the benchmark irrigation rates and the proposed estimates over the five pilot districts. Panels a)-e) show the monthly time series, while a seasonal comparison is provided in panel f). As in the case of the Po valley, the results referring to the non-irrigated season have been masked out and the metrics have been computed by considering irrigation periods only. Very good performances are obtained over three districts, i.e., Coleambally, Murray Mulwala, and Murrumbidgee, across which the *RMSE* varies between 8.65 mm/month and 10.54 mm/month, *r* ranges from 0.66 to 0.84, and the *BIAS* is between -7.26 mm/month and 3.18 mm/month. Even though the timing is properly reproduced,





the SM-based inversion approach overestimates benchmark irrigation over the Murray Wakool site. Finally, unsatisfactory

performances are found over the Western Murray district, where not-negligible underestimates can be observed. Over this

district the SM-based inversion algorithm returns lower estimates with respect to the other test sites and, concurrently,

benchmark rates are higher. The unsatisfactory performances over the Western Murray district could be explained by the

mismatch between the site extent (which is from twenty to sixty times smaller than the areas of the other pilot districts) and

by the adopted spatial resolution, which is a crucial aspect for properly detecting the irrigation signal from space (Dari et al.,

2022a; Massari et al., 2021). The scatter plot provided in panel f) of Figure 9 summarises the long-term comparison. It is

noteworthy that, in this case, the data have been aggregated at the yearly time step by considering April as a starting month,

thus including the irrigation season, which crosses two calendar years. Three representative points are very close to the 1:1

line and the relative error results less than or equal to ± 30% in five cases out of ten.


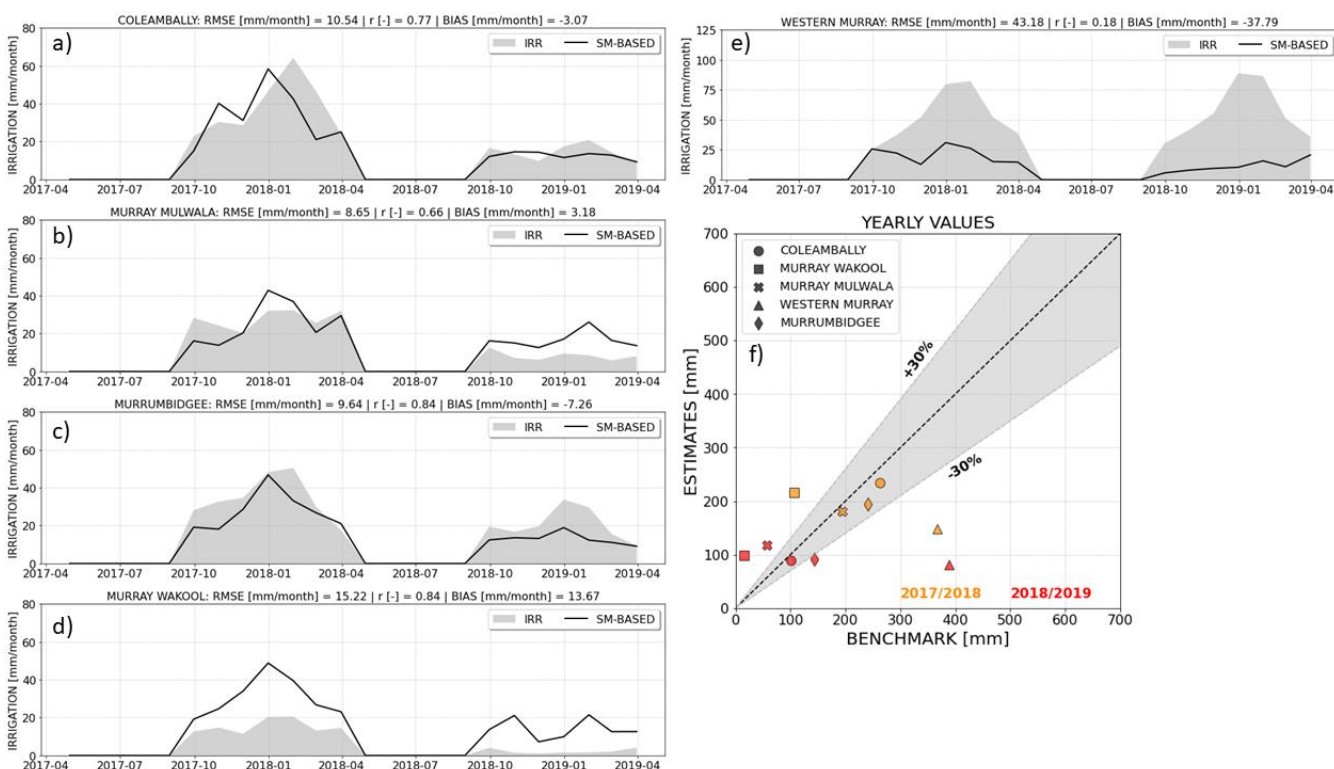

**Figure 9: Panel a)-e): 14-day aggregated time series of irrigation estimates (the black lines) over the five considered pilot districts within the Murray-Darling basin against the benchmark rates (the light grey shaded areas). Panel f): yearly comparison with data referring to different districts and years indicated through different markers and colours, respectively. The interval indicating a**
**relative error of ± 30% is shown as well.**



## 4.4 Limitations

Two main issues may affect the developed data sets, namely irrigation amounts wrongly retrieved over non-irrigated areas and false irrigation rates reproduced during the non-irrigated season. In principle, the detailed knowledge of the actual extent of irrigated areas could allow to estimate how much water is wrongly retrieved where irrigation is not practiced and such an information could be exploited to correct the algorithm output (Jalilvand et al., 2019). Unfortunately, irrigation dynamics are highly uncertain and the actual extent of irrigated areas is not an exception, as demonstrated by the large number of studies aimed at developing remote-sensing-based techniques to map irrigation (see, e.g., Bazzi et al., 2019; Deines et al., 2019; Dari et al., 2021; Elwan et al., 2022). Moreover, the information on the irrigation extent is dynamic in time, as within the areas equipped for irrigation the actually irrigated fields may vary from year to year on the basis of economic factors, climatic conditions and farming strategies (e.g., crop rotation). In summary, the developed data sets could benefit of integration with the spatial information of the actual extent of irrigated areas (when available) or, at least, with global maps of areas equipped for irrigation (e.g., Salmon et al., 2015; Siebert et al., 2015; Nagaraj et al., 2021).

False irrigation rates during non-irrigated seasons represent an additional issue that could affect the proposed data sets. Unlike the irrigation extent, the information on the crop watering period is generally available and more reliable. Moreover, since the irrigation schedule depends on the crop type, cropping calendars can be useful (see e.g., Portmann et al, 2008). Information on the irrigation seasons can be used to postprocess the developed products, as shown here for the Po and Murray-Darling case studies. However, an assessment of retrieved irrigation amounts through the SM-based inversion approach per each month of the year has been carried out. Figure 10 provides the mean of the cumulated irrigation amounts for each month of the years covered by the developed data sets calculated over the pilot districts, i.e., where irrigation surely occurs. The highest-intensity irrigation seasons are highlighted in light green. For the selected sites within the Ebro and the Murray-Darlin basins, the highest rates are retrieved when expected. The same happens for the Po valley, for which the highest peak occurs in June, but a second-not negligible peak can be observed in October as well. Such a circumstance can be attributed to false irrigation rates, as it is not corroborated by significant benchmark irrigation rates in the same period. However, over the portion of the Po valley where the considered pilot districts are located, fruit trees are used to be irrigated even in October and November; hence, a contamination with the signal coming from surrounding pixels where this kind of crop is cultivated could explain the irrigation peak in the autumn season over the Formellino and San Silvestro districts. Nevertheless, the monthly irrigation rates are never equal to zero during the winter period; this is a common issue to all the three case studies and, except for the Ebro basin, it can be considered as a false irrigation, with magnitude over the Po valley higher than over the Murray-Darling basin. These potential errors can be attributed to overestimates of rainfall rates through the proposed approach under humid climate conditions, resulting in water amounts wrongly attributed to irrigation practices.





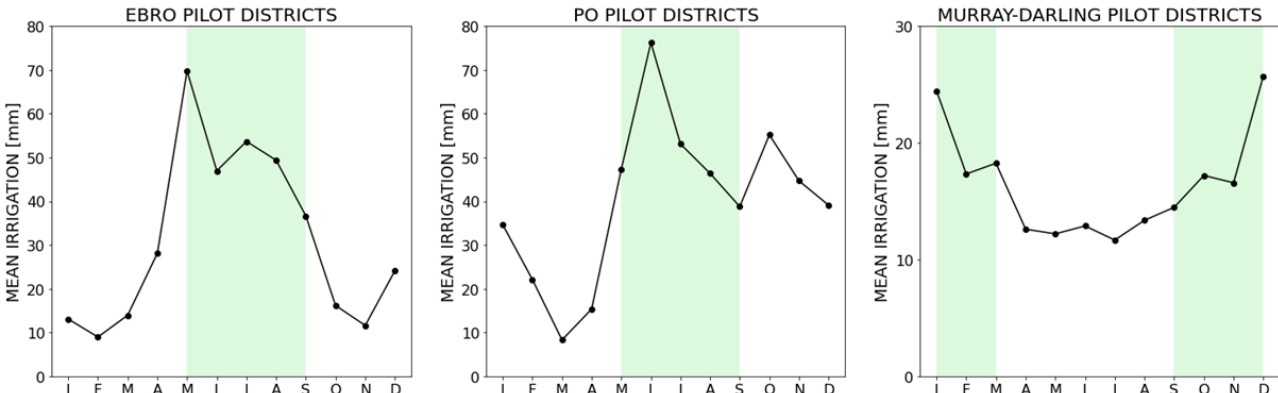

**Figure 10: Mean irrigation amounts retrieved per each month of the year over the pilot irrigation districts of the Ebro basin, the Po valley, and the Murray-Darling basin. Note the different y-axis range for the plot referring to the Murray-Darling case study.**

**4.5 Future plans**

The room for enhancing the retrieved irrigation products largely relies on users' feedback, which are essential to address future developments. In the following, some of the main improvements that could be implemented in the next future are listed:

- Temporal extension of the retrieved irrigation data sets. Currently, the time span for which the data sets have been
produced is constrained by the availability of the input data at the time of products development. For instance, the ending date common to all the data sets is due to the temporal coverage of GLEAM v3.5b.

- Spatial extension of the retrieved irrigation data sets. Undoubtedly, the main challenge is going through a global irrigation product. Nevertheless, the high-resolution (1 km or less) required for the input data sets (at least soil moisture) and the high variability of irrigation practices worldwide makes it necessary for a gradual progression.
Hence, the very imminent challenge is the extension to the country and the continental scales. For instance, the development of irrigation estimates through the SM-based inversion approach over the whole Mediterranean area is a task foreseen within the ESA 4DMED-Hydrology project.

- Exploitation of high-resolution evapotranspiration input data. Along with soil moisture, the evapotranspiration term plays a fundamental role in determining the output of the SM-based inversion approach (Dari et al., 2020). Hence,
the exploitation of higher resolution input PET rates for computing the evapotranspiration term of Eq. (3) (e.g., from a 1 km resolution version of the GLEAM data set over the Mediterranean, which will be developed within the abovementioned 4DMED-Hydrology project) is among the future perspectives of this study.

- Integration with crop calendars and spatial information on actually irrigated areas. As already mentioned in Sub-section 4.4, the postprocessing of the developed data sets with ancillary site-specific information on irrigation extent
and duration is recommended. Hence, users are encouraged to exploit such information, when available, to refine the data. In future, such procedures could be even automatized and implemented into the algorithm. This is already



partly done in the calibration step, in whose first step the potential irrigation days are masked out (see Sub-section 3.1).

## 5 Data availability

The three irrigation data sets presented in this study are freely available at https://doi.org/10.5281/zenodo.7341284.

## 6 Conclusions

This study presents high-resolution irrigation products derived from satellite observations and developed over three highly human-influenced regions. The data sets, developed through the SM-based inversion approach, are one of the main outcomes of the ESA Irrigation+ project. The retrieved irrigation amounts have been validated through a comparison with benchmark 460 rates over selected districts. For the Ebro and the Murray-Darling basins, the amount of the collected information on irrigation dynamics allows a reliable validation, which provides satisfactory results. For the Ebro basin, median values of $RMSE$, $r$, and $BIAS$ equal to 12.4 mm/14-day, 0.66, and -4.62 mm/14-day, respectively, are found. Referring to the Murray-Darling basin, the analogous values are 10.54 mm/month, 0.77, and -3.07 mm/month. The validation over the Po valley is affected by higher uncertainties due to the limited period of in situ irrigation data used as reference, and referring to two very 465 small districts. However, the authors encourage the scientific community to perform deeper validation studies, as the main aim of this work is the use of the developed products. Under this perspective, limitations and suggested postprocessing strategies are highlighted in Sub-section 4.4. The presented irrigation products are the first regional-scale gridded data sets retrieved from satellite observation at a spatial resolution suitable for the water resource management in agriculture. Hence, this kind of applications are fostered, as for instance the ingestion of the developed datasets in systems providing irrigation 470 advice or performing irrigation water accounting. Of course, the ingestion of the proposed irrigation data sets into hydrological and land surface modeling is possible as well. Users' feedback will be essential to address future implementations, with the final aim of building an operational system for high-resolution irrigation water monitoring from space.

## Author contribution

JD: Conceptualization, Methodology, Software, Validation, Formal analysis, Investigation, Writing – original draft preparation, Writing - review & editing, Visualization. LB: Conceptualization, Methodology, Investigation, Writing - review & editing, Supervision. SM: Formal analysis, Investigation, Validation, Writing – review & editing. CM: Investigation, Writing - review & editing, Supervision. AT: Writing - review & editing, Supervision. AT: Investigation, Writing - review & editing, Supervision. SB: Investigation, Writing - review & editing, Supervision. RQ: Resources, Writing - review & editing.



MV: Resources, Writing - review & editing. VF: Resources, Writing - review & editing. ABO: Resources, Writing - review & editing. PQS: Investigation, Resources, Writing - review & editing. DB: Resources, Writing - review & editing. EV: Writing - review & editing, Supervision.

**Competing interest**

The authors declare that they have no conflict of interest.

**Acknowledgements**

The authors acknowledge the support from the European Space Agency (ESA) under the IRRIGATION+ project (contract n. 4000129870/20/I-NB); for further details please visit https://esairrigationplus.org/. The authors wish to thank the SAIH (*Sistema Automático de Información Hidrológica*) Ebro and CER (*Canale Emiliano Romagnolo*) for providing the irrigation benchmark values for the Ebro basin and Po valley pilot districts, respectively. The authors would like to thank the Irrigation
Infrastructure Operators (IIOs) that provided the Australian irrigation benchmark data for this study.

**Appendix A. Calibrated parameters**

In this section, the outcomes of the calibration procedure described in Sub-section 3.1 and summarised in Figure 2 are provided. For the parameters $a$, $b$, and $Z^*$, a spatially distributed calibration has been performed. The calibrated values of such parameters for the Ebro basin are shown in the maps of Figure A.1 and in the boxplots of Figure A.2. Across the whole
study area, the median values of $a$, $b$, and $Z^*$ are equal to 18.84 mm, 3.98, and 79.82 mm, respectively. The calibrated value of the $F$ parameter (adopted over the whole domain) is equal to 1.37.





**Figure A.1: Spatial distribution of the calibrated values of the parameters $a$, $b$, and $Z^*$ over the Ebro basin.**


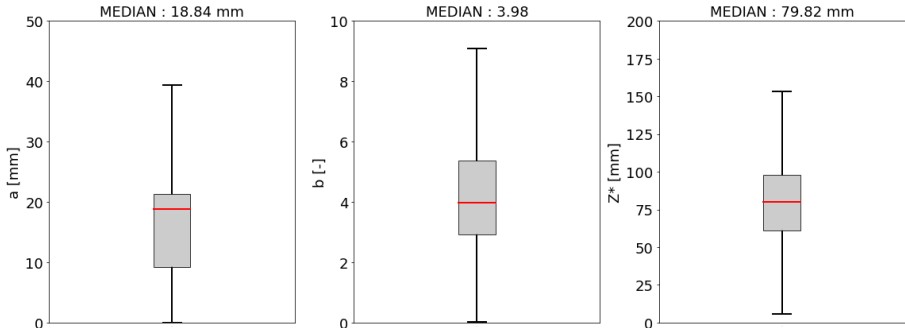

**Figure A.2: Boxplot showing the calibrated values of the parameters $a$, $b$, and $Z^*$ over the Ebro basin.**



The calibrated values of the parameters $a$, $b$, and $Z^*$ obtained over the Po valley are shown in the maps of Figure A.3 and in the boxplots of Figure A.4. The median values of $a$, $b$, and $Z^*$ resulting from the calibration step are equal to 7.02 mm, 1.40, and 97.63 mm, respectively. The calibrated value of the $F$ parameter (adopted over the whole domain) is equal to 0.60.


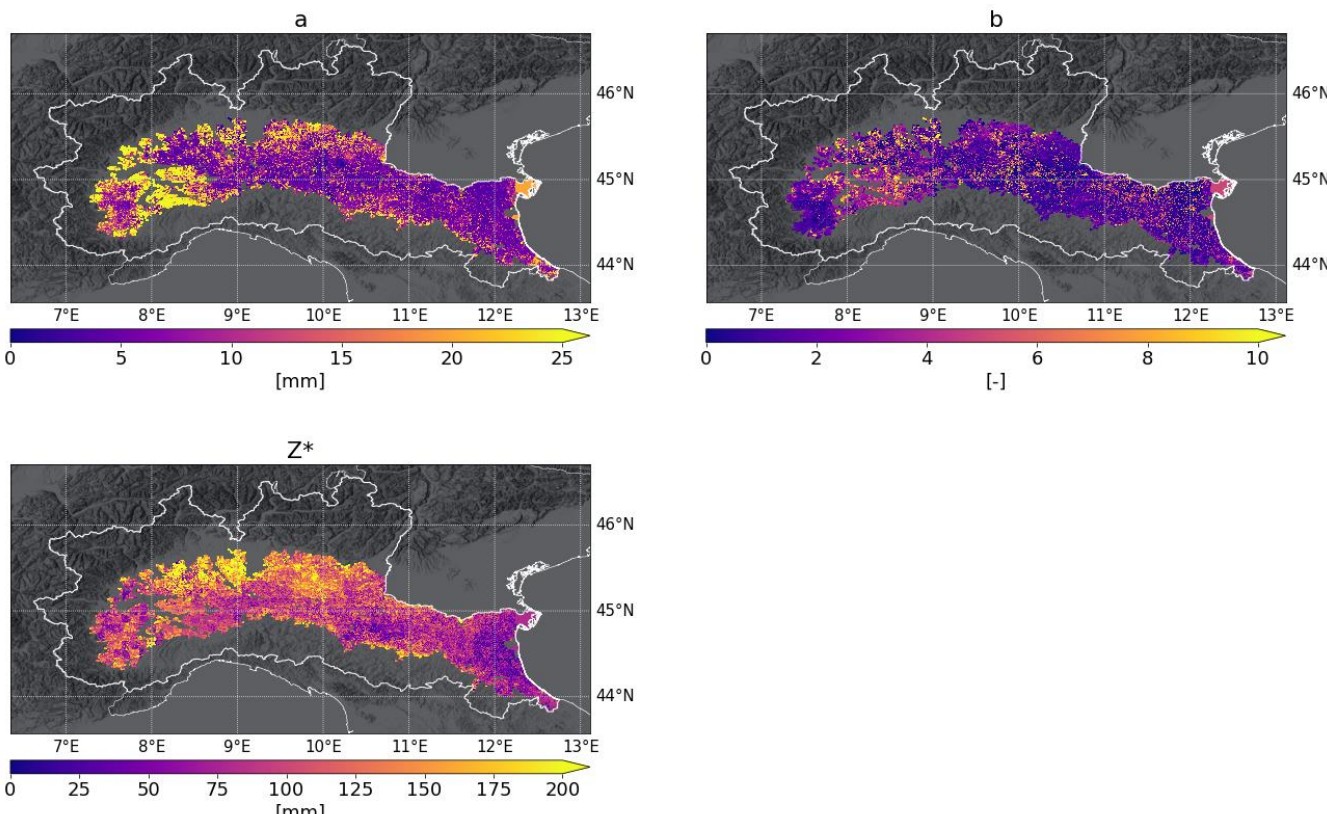

Figure A.3: Spatial distribution of the calibrated values of the parameters $a$, $b$, and $Z^*$ over the Po valley.

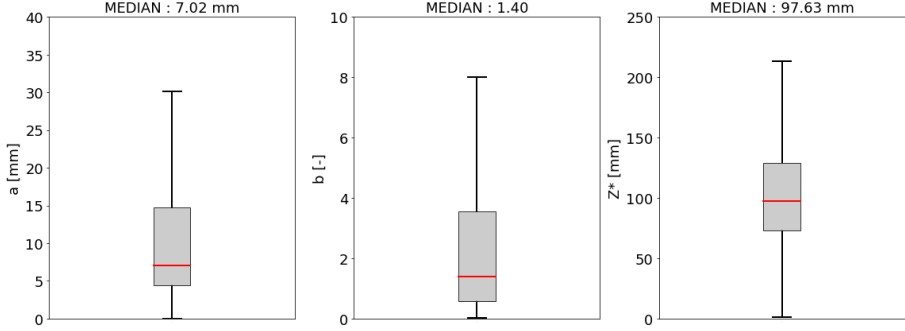


Figure A.4: Boxplot showing the calibrated values of the parameters $a$, $b$, and $Z^*$ over the Po valley.



Finally, the calibrated values of the parameters $a$, $b$, and $Z^*$ obtained over the Murray-Darling basin are provided in the maps of Figure A.5 and in the boxplots of Figure A.6. The median values of $a$, $b$, and $Z^*$ result equal to 2.32 mm, 8.27, and 50.21 mm, respectively. The calibrated value of the $F$ parameter (adopted over the whole domain) is equal to 0.60.


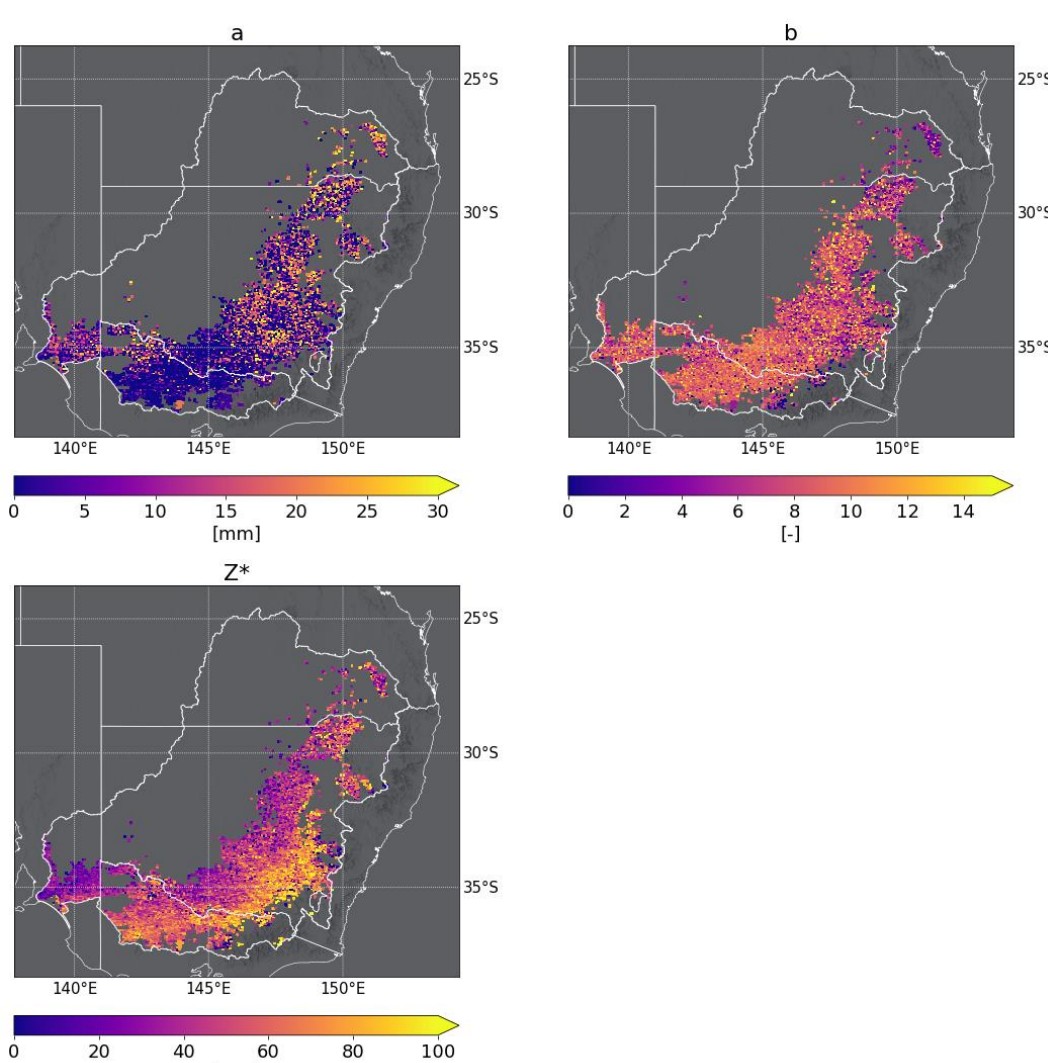

**Figure A.5: Spatial distribution of the calibrated values of the parameters $a$, $b$, and $Z^*$ over the Murray-Darling basin.**



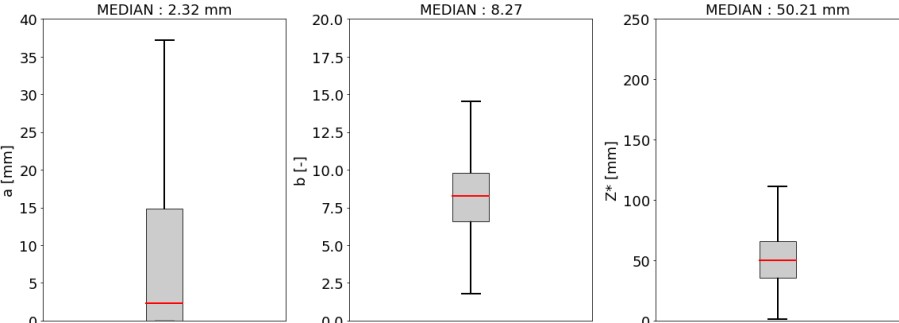

**Figure A.6: Boxplot showing the calibrated values of the parameters $a$, $b$, and $Z^*$ over the Murray-Darling basin.**

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
