# Peer review of "Regional data sets of high-resolution (1 and 6 km) irrigation estimates from space"

_Earth System Science Data, 2022_

## Referee Comment (RC1)

Reviewer's report for essd-2022-403

Regional data sets of high-resolution (1 and 6 km) irrigation estimates from space

Summary -
Irrigation water has been a majority portion of human total water use but detailed information at high-resolution for water resource instructions has been limited. In this study, the authors presented high-resolution regional datasets derived from remote sensing through SM-based method (water balance equation) for three basins, Erbo and Po basin (1-km) in Europe, and Murray basin (6-km) in Australia. The products are evaluated through detailed benchmark irrigation rate data collected in these three basins. The results are valid and valuable for Erbo and Murray basin, as for Po basin, large uncertainty exist due to limited length of benchmark data. The authors highlighted the limitations of this method and proposed ideas for future studies.
The manuscript presented has great significance for remote sensing and land model communities and has great potential to achieve large-scale application. Despite its importance, some comments and issues need to be addressed before it can be accepted for the journal. Please see my comments and questions below:

General comments -

1. Eq (1) shows the water balance terms used in the SM-based method. Does any of these three basin would have snowfall and snowpack in winter and snowmelt contributing to soil moisture water balance in spring? As no word of snow is mentioned in the manuscript, would it be possible that the overestimated high irrigation rate in winter and early spring peak due to a missing (snow) component in Eq (1).

2. Another possibly neglected component would be the groundwater contribution to soil moisture, the $g(t)$ term, if the groundwater table is shallow, it would be possible that rainfall from other area could drain to river and groundwater aquifer which provides lateral transport of moisture, contributing to soil moisture.

3. 3.2 Input data sets, the authors used different sources of input data sets with various resolution, soil moisture (1-km), PET (0.25°) and precipitation (9-km), can the authors also provide a discussion on the way of spatial aggregation, would it affect the results obtained in the dataset?

4. The authors also highlighted the uncertainty and future exploitation of ET data, would it be possible to apply a physical process-based land surface model (LSM), which has more sophisticated ET calculation, to obtain SM-based irrigation rate, rather than the soil moisture balance Eq (1)?

5. The scatter plot in Figure 5,7,9. It seems that there are some systematic underestimation in Ebro basin and Murray baxin, and overestimation in Po basin. Any speculation on these, would it be due to the physical landscape, i.e. missing processes in Eq (1) used to estimate SM? If this is true, what caution would the authors recommend to users when propogating this method to use in large regional application in other regions of the world?

6. Do these three basins in pilot areas use the same irrigation method? If different irrigation methods are used in these three basins, would these affect the SM-based inversion approach, reflected on the results? The authors could provide a discussion on the irrigation methods and also

7. The authors mentioned applying irrigation map and crop calendar to constrain irrigation dataset in Discussion. When applying these constrains, would it affect the calibration parameters, i.e. these parameters would need to re-calibrate?

Specific comments -
1. In general, the presentation of the manuscript is good. But there are several places where paragraphs are too long. For example, the first paragraph of Introduction is too long, the authors may divide it into separate paragraphs that may be helpful for readers.
2. L307: "… to the Urgell district" could use a separate paragraph.

---

## Referee Comment (RC2)

**General comments:**

This study presents gridded irrigation datasets for three major irrigated agriculture regions, including two sites in Europe and one site in Australia. The SM-based inversion water balance model is utilized to estimate irrigation using satellite soil moisture data. Given the scarcity of irrigation data at a large scale, this dataset has a significant potential to serve as a reference for other studies and in this sense, it is a good fit for ESSD. However, it is imperative that the uncertainties involved in the production of this dataset are clearly outlined in the text, particularly in the abstract and conclusion. Additionally, it should be clarified whether this data can be used to validate other studies or if it requires further validation with in-situ datasets (as mentioned in L465 of the conclusion). I outlined my comments below, addressing these comments the paper can be considered for publication:

**Specific comments:**

**Major**

1. The reliability of the irrigation dataset outside of the benchmark irrigated areas is questionable as the model is calibrated and validated over the same sites. In order to have a better understanding of the model error outside of the benchmark area it is better to keep at least one of the benchmark areas in each region just for validation (not use it in the calibration process). This way, it will be easier to see how well the model performs in areas that are not part of the benchmarked sites, which constitute more than 80% of total case study area.

2. It is not clear whether this product is produced as a reference (benchmark) for other studies (as it can be inferred from the title or L19-20) or if the authors ask the community to validate this product (L108-110, L465, L471). This should be explicitly mentioned in the abstract.

3. L195: This suggests that you calibrate F over a small fraction of irrigated area, and use the same value for the whole region while they might have different crop types and surface characteristics. Please discuss how this can affect the estimated irrigation, especially over the white area of figure 3.

**Moderate**

1. L86: I guess it is not improving rather it is an estimation as there is no irrigation module involved in their method, in this sense it is different from the rest of DA studies mentioned here which are improving irrigation estimated by an irrigation module.

2. Figure1: I prefer the climate description over the abbreviations as the readers can better relate to it, for example:

   *Cfa* = Humid subtropical climate

   To save some space, you can use one legend for both maps if they are identical
3. L127-128: The sentences are sometimes unnecessarily complex and long, use plain language and shorter sentences.
4. L173: Are negative irrigation rates significantly lower than the positive irrigation rates, if yes please report this in your result section.
5. L183: Do you have supplemental irrigation in any of these regions? Please discuss this in the manuscript.
6. L184-186: Some sentences like this one are long you can break them into shorter sentences that are easier to follow by the readers.
7. L186: Using insitu irrigation data in the calibration process can create some limitations as this data is not always available or it might be provided for a part of the study area (like this study). Have you tried to calibrate the F parameter during the rainy season for instance during the drydowns after the rainfall, in this case, you will have the F value for all the pixels and not just in pixels for which you have irrigation data. Another suggestion is to calibrate the model over the rainfed cropland areas during the irrigation season. It would be really interesting to see how much skill will be gained/lost for irrigation estimation.
8. L235: What is the temporal resolution of GNSS soil moisture? It is expected to be more frequent, which I think is an advantage for the SM-based inversion algorithm. You can highlight it here.
9. One of the important inputs (or the most important one!) for your approach is soil moisture data. You have used different satellite technique to retrieve soil moisture that has different accuracy, especially since the GNSS-based method tends to be less accurate. Consequently, the estimated irrigation should not be treated the same, this should be highlighted in the manuscript.
10. are the benchmark irrigation districts representative of the larger irrigated regions (white area in Figure 3)?
11. Table. 1. Do you have one daily irrigation volume value for an 887 km2 area in the Urgell site? Or you have multiple irrigation measurement stations at each benchmark site that measure irrigation for smaller agricultural land parcels? This needs to be clearly explained in the manuscript.
12. Which part of the data is used for calibration and which part is used for validation, add two columns to table one and provide this information for each site.

**Minor**

1. L94: cannot make sense of this line: "*The high-resolution retrievals of the latest satellite capabilities open …*" rephrase, please.
2. L116: what do you mean by interesting in this sentence: "*with oceanic climate areas (Cfb) **interesting** the upper part…*"
3. The sentence at L125-126 is difficult to follow
4. L184: *This first step involves* calibration of…?

5. L188: the second **and** is extra: *… the values of $a$, $b$, and $Z*$ **and** are re-calibrated…*
6. L248: I think it is better to use irrigation depth or height instead of "irrigation doses"

---

## Author Comment (AC1)

**Response to the reviewers' comments**

The authors of this study wish to thank the reviewers for their accurate and useful comments. We believe that the expert reviewers' opinions will provide us guidance to improve the manuscript. A detailed response is provided in the following. In black, we have reported the reviewers' comments, in red, the detailed replies, and in blue, the sentences that will be changed and/or added in the revised manuscript to address reviewers' comments.

**Reviewer #1:** Summary -

Irrigation water has been a majority portion of human total water use but detailed information at high-resolution for water resource instructions has been limited. In this study, the authors presented high-resolution regional datasets derived from remote sensing through SM-based method (water balance equation) for three basins, Erbo and Po basin (1-km) in Europe, and Murray basin (6-km) in Australia. The products are evaluated through detailed benchmark irrigation rate data collected in these three basins. The results are valid and valuable for Erbo and Murray basin, as for Po basin, large uncertainty exist due to limited length of benchmark data. The authors highlighted the limitations of this method and proposed ideas for future studies. The manuscript presented has great significance for remote sensing and land model communities and has great potential to achieve large-scale application. Despite its importance, some comments and issues need to be addressed before it can be accepted for the journal. Please see my comments and questions below:

We wish to thank the reviewer for appreciating our work and for the useful suggestions provided.

General comments -

1. Eq (1) shows the water balance terms used in the SM-based method. Does any of these three basin would have snowfall and snowpack in winter and snowmelt contributing to soil moisture water balance in spring? As no word of snow is mentioned in the manuscript, would it be possible that the overestimated high irrigation rate in winter and early spring peak due to a missing (snow) component in Eq (1).

We thank the reviewer for this interesting comment. Actually, the source for rainfall input we are using is the total precipitation from the ERA5-Land data set that involves snow as well. Nevertheless, the contribution of snowfall (if any) is definitely not conclusive, as for each basin the algorithm is run over flat agricultural valleys only. For similar reasons, as explained in the manuscript at lines 164-165, the surface runoff is neglected and the assumption is valid for runoff generated by snowmelt as well.

2. Another possibly neglected component would be the groundwater contribution to soil moisture, the g(t) term, if the groundwater table is shallow, it would be possible that rainfall from other area could drain to river and groundwater aquifer which provides lateral transport of moisture, contributing to soil moisture.

We thank the reviewer for the suggestion. We wish to highlight that in our approach the g(t) term represents the amount of water exiting from the bottom of the soil layer. Hence, it is a term quantifying groundwater recharge, not considering potential groundwater contribution to soil moisture. It is important to underline that in case of remote sensing soil moisture the information refers to the soil surface (generally less than 5 cm). Hence, soil moisture alterations due to shallow groundwater may occur very close to river reaches and/or deltas only, namely areas that are masked out. Hypothetical border effects over pixels adjacent to water bodies (if any) are expected to be negligible.

3. 3.2 Input data sets, the authors used different sources of input data sets with various resolution, soil moisture (1-km), PET (0.25°) and precipitation (9-km), can the authors also provide a discussion on the way of spatial aggregation, would it affect the results obtained in the dataset?

We thank the reviewer for this comment, however, believe that the spatial resolution of the ERA5-Land reanalysis used to derive rainfall rates (~9 km) does not represent an important limitation for the proposed methodology. Conversely, the spatial resolution of the PET term surely has a more important role in determining the algorithm output, considering also that the product currently used for PET rates is characterized by a spatial resolution pretty coarser with respect to the target of this application. Hence, we will further specify the importance of future implementations using higher resolution PET rates from GLEAM (which were not available at the time of our implementations) at the third point of the "Future plans" section (line 443), which will read as follows:

"Exploitation of high-resolution evapotranspiration input data. Along with soil moisture, the evapotranspiration term plays a fundamental role in determining the output of the SM-based inversion approach (Dari et al., 2020). Hence, the exploitation of evapotranspiration estimates at a spatial resolution matching the scale at which irrigation occurs is expected to bring benefits to the outcomes of the SM-based inversion approach. For this reason, the use of higher resolution input PET rates for computing the evapotranspiration term of Eq. (3) (e.g., from a 1 km resolution version of the GLEAM data set over the Mediterranean, which will be developed within the abovementioned 4DMED-Hydrology project) is among the future perspectives of this study."

We wish to highlight that the implementation of the methodology with 1 km PET rates from GLEAM is a process already running over the two European basins that will likely lead to a version 2 of the irrigation data sets for the Ebro basin and the Po valley.

4. The authors also highlighted the uncertainty and future exploitation of ET data, would it be possible to apply a physical process-based land surface model (LSM), which has more sophisticated ET calculation, to obtain SM-based irrigation rate, rather than the soil moisture balance Eq (1)?

The evapotranspiration term of Eq (1) relies on potential evapotranspiration rates, PET, which can be derived from different sources, involving the output of LSMs. In previous works, the method was implemented with evapotranspiration data coming from several sources and modeling approaches (see, e.g., Dari et al., 2022). Nevertheless, we believe that maximizing the use of data sets relying also on observations is an added value for our purpose, as irrigation is a human-induced process which is usually missed or poorly parameterized by models.

5. The scatter plot in Figure 5,7,9. It seems that there are some systematic underestimation in Ebro basin and Murray baxin, and overestimation in Po basin. Any speculation on these, would it be due to the physical landscape, i.e. missing processes in Eq (1) used to estimate SM? If this is true, what caution would the authors recommend to users when propogating this method to use in large regional application in other regions of the world?

In our opinion, the main reason for the different performances obtained over the Po valley as compared to the other two pilot areas is attributable to the climatic features. In fact, systematic underestimations over the pilot areas characterized by an arid or semi-arid climate (water-limited regimes) are obtained independently on the soil moisture product used (RT1 Sentinel-1 for the Ebro basin and CYGNSS for the Murray-Darling basin). Conversely, irrigation overestimates over the Po valley can be attributed to rainfall overestimates in a humid context, as specified at lines 424-425 of the manuscript. However, given the high novelty degree of the research topic and the very scarce knowledge about irrigation dynamics worldwide (and consequently scarce availability of benchmark data for validating the proposed estimates) we prefer not speculating too much on the aspects mentioned by the reviewer, but we encourage the scientific community in checking, testing, and validating the developed products. We think that potential users are properly informed about the caution to be adopted by reading the "Limitations" section (lines 396-425 of the manuscript).

6. Do these three basins in pilot areas use the same irrigation method? If different irrigation methods are used in these three basins, would these affect the SM-based inversion approach, reflected on the results? The authors could provide a discussion on the irrigation methods and also

7. The authors mentioned applying irrigation map and crop calendar to constrain irrigation dataset in Discussion. When applying these constrains, would it affect the calibration parameters, i.e. these parameters would need to re-calibrate?

We merge the responses to points 6 and 7. The reviewer is right, different irrigation techniques can be differently detected by satellite soil moisture and, in turn, lead to different performances in estimating water amounts through the SM-based inversion approach. Surely, different techniques are adopted across basins and within each basin. In order to clarify potential issues due to the adoption of different irrigation techniques, we will add the following sentence referring to studies in which this issue is deepened at line 356.

"Previous studies highlighted how the irrigation method affects the capability of remotely sensed soil moisture products to detect irrigation-driven changes (see, e.g., Gao et al., 2018; Dari et al., 2021)."

The following reference will be added in the proper section:

Gao, Q., Zribi, M., Escorihela, M.J., Baghdadi, N., and Quintana-Seguí, P.: Irrigation mapping using Sentinel-1 time series at field scale, Remote Sens., 10, 1495, https://doi.org/10.3390/rs10091495, 2018.

Regarding the spatial and temporal constraints for refining the irrigation estimates, no, there is no need to re-calibrate the algorithm parameters. What we suggest is a postprocessing of the results, as specified at lines 448-453 of the manuscript.

Specific comments -

1. In general, the presentation of the manuscript is good. But there are several places where paragraphs are too long. For example, the first paragraph of Introduction is too long, the authors may divide it into separate paragraphs that may be helpful for readers.
   We will separate the first paragraph of the introduction as suggested by the reviewer.
2. L307: "… to the Urgell district" could use a separate paragraph
   We will separate the paragraph as suggested by the reviewer.

**References:**

Dari, J., Quintana-Seguí, P., Morbidelli, R., Saltalippi, C., Flammini, A., Giugliarelli, E., Escorihuela, M.J., Stefan, V., and Brocca, L.: Irrigation estimates from space: Implementation of different approaches to model the evapotranspiration contribution within a soil-moisture-based inversion algorithm, Agric. Water Manag., 265, 107537, https://doi.org/10.1016/j.agwat.2022.107537, 2022.

---

## Author Comment (AC2)

**Response to the reviewers' comments**

The authors of this study wish to thank the reviewers for their accurate and useful comments. We believe that the expert reviewers' opinions will provide us guidance to improve the manuscript. A detailed response is provided in the following. In black, we have reported the reviewers' comments, in red, the detailed replies, and in blue, the sentences that will be changed and/or added in the revised manuscript to address reviewers' comments.

**Reviewer #2:**

General comments:

This study presents gridded irrigation datasets for three major irrigated agriculture regions, including two sites in Europe and one site in Australia. The SM-based inversion water balance model is utilized to estimate irrigation using satellite soil moisture data. Given the scarcity of irrigation data at a large scale, this dataset has a significant potential to serve as a reference for other studies and in this sense, it is a good fit for ESSD. However, it is imperative that the uncertainties involved in the production of this dataset are clearly outlined in the text, particularly in the abstract and conclusion. Additionally, it should be clarified whether this data can be used to validate other studies or if it requires further validation with in-situ datasets (as mentioned in L465 of the conclusion). I outlined my comments below, addressing these comments the paper can be considered for publication:

We thank the reviewer for the effort put in suggesting improvements for our manuscript.

Specific comments:

Major

1. The reliability of the irrigation dataset outside of the benchmark irrigated areas is questionable as the model is calibrated and validated over the same sites. In order to have a better understanding of the model error outside of the benchmark area it is better to keep at least one of the benchmark areas in each region just for validation (not use it in the calibration process). This way, it will be easier to see how well the model performs in areas that are not part of the benchmarked sites, which constitute more than 80% of total case study area.

We thank the reviewer for this comment. However, we don't think the reliability of the results is affected by calibration and validation over the same sites. In fact, for the Ebro and the Murray-Darling basins the total time span for which the data sets have been developed has been split into a first sub-period of calibration and a second one for validation. Moreover, the calibration of the F parameter has been carried out considering only part of the irrigation districts for which benchmark irrigation rates have been collected. The only exception is the Po valley, for which the very limited information available there constrained adopting the same time span for calibration and validation. It is important to highlight that the calibration process is iterative; hence, validation results are never fully overlapping with those coming from the calibration step, since three of the four algorithm parameters are re-tuned. Furthermore, we would like to point out that just one parameter (i.e., F) is calibrated using reference irrigation amounts, while the other three algorithm's parameters are calibrated against rainfall rates, which are widely available.

We will further overcome this issue as we managed to extend until 2020 the time series of benchmark irrigation rates for the two pilot districts within the Po valley, thus making it possible, also for this case study, to split the whole simulation period into a first half of calibration and a second half of validation. Under this perspective, results shown in Figure 7 will be updated.

2. It is not clear whether this product is produced as a reference (benchmark) for other studies (as it can be inferred from the title or L19-20) or if the authors ask the community to validate this product (L108-110, L465, L471). This should be explicitly mentioned in the abstract.

The products presented in this work are the first irrigation data sets derived from satellite observations at high resolution and covering wide areas (regional scale). Obviously, when developing a data set the final aim is that data are usable in other studies and this case is not an exception. Notwithstanding, given the high novelty level of the research topic together with the scarce availability of benchmark irrigation data for comparison, we believe that validation from the scientific community, eventually owning additional in situ data is a necessary step. We will clarify this aspect by adding the following sentence at the end of the abstract:

"The developed products are made available to the scientific community for use and further validation."

3. L195: This suggests that you calibrate F over a small fraction of irrigated area, and use the same value for the whole region while they might have different crop types and surface characteristics. Please discuss how this can affect the estimated irrigation, especially over the white area of figure 3.

The reviewer is perfectly right, this issue is exactly the reason for which the other three parameters are re-tuned in a second step. In other words, once the optimal F value is fixed after calibration against the sum of rainfall and irrigation, the parameters a, b, and Z* are adjusted to compensate for the effect of adopting over rainfed agricultural areas a value of the F parameter, calibrated where irrigation occurs. In order to further clarify this point, the following sentence will be added at line 189:

"In this way, the effects of adopting the same $F$ value for rainfed agricultural areas, calibrated where irrigation occurs, are compensated."

Moderate

1. L86: I guess it is not improving rather it is an estimation as there is no irrigation module involved in their method, in this sense it is different from the rest of DA studies mentioned here which are improving irrigation estimated by an irrigation module.

We will change the sentence as follows:

"In this context, Abolafia-Rosenzweig et al. (2019) designed a particle batch smoother DA system for remote-sensing-based soil moisture assimilation into the VIC (Variable Infiltration Capacity) model (Liang et al., 1994) to estimate irrigation amounts."

2. Figure1: I prefer the climate description over the abbreviations as the readers can better relate to it, for example:

Cfa = Humid subtropical climate

To save some space, you can use one legend for both maps if they are identical

We will modify the figure according to the reviewer's suggestion. Actually, even using one legend only, a description for 30 classes should be added, i.e., a lot of text in the figure. Hence, we will evaluate a couple of strategies: (i) to report in the legend the classes present in the study are only or (ii) to keep the legend as it is and specify the description of the classes detectable over the pilot basins in the figure caption.

3. L127-128: The sentences are sometimes unnecessarily complex and long, use plain language and shorter sentences.

We will revise the sentence as follows:

"The Po valley is part of the Po river basin, the longest river in Italy. It is a critical hotspot for studying the impact of human activities on the water cycle, considering the key role of the agricultural production and the increasing frequency of severe drought events over this area (Strosser et al., 2012; Ceppi et al., 2014; Formetta et al., 2022)."

4. L173: Are negative irrigation rates significantly lower than the positive irrigation rates, if yes please report this in your result section.

Negative irrigation rates retrieved by the SM-based inversion approach are significantly lower than positive ones, as demonstrated in previous studies describing former versions of the method. For instance, Jalilvand et al. (2019) applied the methodology with coarse resolution soil moisture as input finding a mean rate of negative irrigation rates lower than one third of mean irrigation amounts retrieved. A reference to the mentioned work, which is already included in the reference section, will be added at line 173 of the manuscript.

5. L183: Do you have supplemental irrigation in any of these regions? Please discuss this in the manuscript.

Supplemental irrigation is practiced in some portions of the study areas, e.g., in the Ebro basin such a technique is sometimes used for vineyards. However, this irrigation method is not adopted in any of the pilot districts, where irrigation is aimed at the maximum productivity. This is valid for all the three regions considered. Hence, supplemental irrigation cannot affect the validation results shown in Figures 5, 7, and 9. On this basis, we believe there is no need to mention this aspect in the manuscript.

6. L184-186: Some sentences like this one are long you can break them into shorter sentences that are easier to follow by the readers.

The sentence will be divided into three shorter ones:

"This first step involves the calibration of a, b, and Z* parameters that are optimized by minimizing the Root Mean Square Difference (RMSD) against reference rainfall rates. During this phase, implemented for each pixel, the evapotranspiration adjustment factor, F, is assumed equal to 1. Then, F is calibrated against the sum of rainfall plus irrigation over selected pilot sites where irrigation rates are known."

7. L186: Using insitu irrigation data in the calibration process can create some limitations as this data is not always available or it might be provided for a part of the study area (like this study). Have you tried to calibrate the F parameter during the rainy season for instance during the drydowns after the rainfall, in this case, you will have the F value for all the pixels and not just in pixels for which you have irrigation data. Another suggestion is to calibrate the model over the rainfed cropland areas during the irrigation season. It would be really interesting to see how much skill will be gained/lost for irrigation estimation.

We agree with the reviewer, the scarce availability of irrigation rates worldwide can represent a limit, as this kind of data is needed to calibrate the F parameter. Once again, we recall that only this parameter is calibrated using reference irrigation amounts, while the other three algorithm's parameters are calibrated against rainfall rates, which are widely available. The calibration procedure currently implemented is the best compromise between reliable results and an as much as possible generalized configuration. During the method evolution so far, we tried what suggested by the reviewer and much more. For instance, the calibration over rainfed cropland has been implemented in a couple of previous works (Dari et al., 2020; Dari et al., 2022). Nevertheless, this configuration is site-specific and not easy to extend to wider scales. Also, this approach can lead to underestimates of the evapotranspiration over irrigated fields (where the actual rate can be even higher than the potential one). In the Irrigation+ project, the framework within which the presented irrigation data sets have been developed, different calibration configurations have been tested and the performances have been assessed through preliminary analysis carried out with multiple soil moisture products over 100+ pilot sites located over 6 test areas. More in detail, tests in which the algorithm

has been calibrated by considering rainfall only as a benchmark (either considering rainy seasons only or not) or rainfall plus irrigation (when available) have been carried out. For each configuration, different data sources for the evapotranspiration term have been tested. As an example, Figure R.1 summarizes performances (Pearson correlation, r, BIAS, and RMSE) obtained over the Australian pilot districts obtained by calibrating against rainfall plus irrigation (panel a) or against rainfall only (panel b). Results refer to experiments carried out by considering soil moisture data from different products, namely CCI ACTIVE, SMAP, and two CYGNSS versions. A similar example but referring to the Po valley can be found in (Modanesi et al., 2021). All the analyses mentioned here represent work carried out within a research project in order to develop the most reliable configuration possible to produce irrigation data sets to be delivered to the scientific community. The final result is the methodology described in this study.

[Figure]

Figure R.1. Performance assessment over the Australian sites by using different soil moisture products and by calibrating the algorithm's parameters against rainfall plus irrigation (panel a) and against rainfall only (panel b).

8. L235: What is the temporal resolution of GNSS soil moisture? It is expected to be more frequent, which I think is an advantage for the SM-based inversion algorithm. You can highlight it here.

9. One of the important inputs (or the most important one!) for your approach is soil moisture data. You have used different satellite technique to retrieve soil moisture that has different accuracy, especially since the GNSS-based method tends to be less accurate. Consequently, the estimated irrigation should not be treated the same, this should be highlighted in the manuscript.

We merge the responses to points 8 and 9. We really appreciate these two comments and we agree with both of them. In fact, a higher temporal resolution surely brings benefits in the outcomes of the SM-based inversion approach. Nevertheless, the spatial resolution is an important aspect as well and, in this case, the

RT1 Sentinel-1 data set brings another crucial advantage with respect to CYGNSS, especially considering the target of this work. Indeed, irrigation detection needs both high temporal and spatial resolution (i.e., in order to detect local irrigation application), however a trade-off between them is needed given the current satellite missions' scenario. We are aware that irrigation estimates relying on different soil moisture products are affected by different levels of uncertainties but our aim is not focused on comparing the performances of different products over the same area. As already pointed out in the previous response (n. 7), the algorithm settings implemented in this study (involving also the choice of the input data sets) are the trade-off between data availability (e.g. Sentinel-1 over Europe has a higher temporal coverage with respect to Australia) and the best performances obtained among several exploratory experiments carried out within the Irrigation+ project. Hence, we would prefer not emphasizing too much comparisons among input data sets and irrigation estimates retrieved over different areas. However, we think we can remark that the developed products rely on different soil moisture product by adding the following sentence at line 463 of the manuscript:

"Note that the two above mentioned data sets are based on different soil moisture products."

10. are the benchmark irrigation districts representative of the larger irrigated regions (white area in Figure 3)?

Yes, the selected districts are representative of the irrigated regions, especially in terms of the scale at which irrigation is practised within the considered areas. Just to clarify, the white areas in Figure 3 represent all the agricultural areas over which the algorithm has been implemented (as specified at lines 178-181), which includes both irrigated and rainfed areas.

11. Table. 1. Do you have one daily irrigation volume value for an 887 km2 area in the Urgell site? Or you have multiple irrigation measurement stations at each benchmark site that measure irrigation for smaller agricultural land parcels? This needs to be clearly explained in the manuscript.

No, for the Ebro basin we don't have data referring to agricultural parcels, but data referring to water volumes flowing through the canals feeding each district. Except for the Urgell, all the pilot districts are fed by one canal. The Urgell area is fed by two canals. We will specify this by adding the following sentence at line 248:

"The Urgell district is fed by two canals, while the others receive water from one canal each."

12. Which part of the data is used for calibration and which part is used for validation, add two columns to table one and provide this information for each site.

We really thank the reviewer for the suggestion, but we think that adding such information in Table 1 could bring some confusion. In fact, only data referring to part of the total pilot districts have been used for calibration purposes. Moreover, the iterative structure of the calibration approach makes validation and calibration outcomes not superimposable during the calibration time window. We believe that the information required by the reviewer is thoroughly provided at lines 268-273. We wish to point out that the sentence referring to the Po case study will be changed since longer time series of benchmark irrigation rates have been collected. Similarly, results shown in Figure 7 will be updated.

Minor

1. L94: cannot make sense of this line: "The high-resolution retrievals of the latest satellite capabilities open …" rephrase, please.

The sentence will be rephrased as follows:

"The high-resolution observations of the most recent satellite technologies open unprecedented perspectives in the irrigation quantification activity".

2. L116: what do you mean by interesting in this sentence: "with oceanic climate areas (Cfb) interesting the upper part…"

The sentence will be modified as follows:

"…but in its upper part there are also areas with oceanic climate (Cfb)".

3. The sentence at L125-126 is difficult to follow

We will rephrase the sentence as follows:

"In order to satisfy the average agricultural annual demand, quantified in 7623 hm3/year, an area of almost 9660 km2 is conceded for irrigated surface over the basin."

4. L184: This first step involves calibration of…?

The sentence will be modified as suggested by the reviewer.

5. L188: the second and is extra: … the values of $a$, $b$, and $Z*$ and are re-calibrated…

We thank the reviewer. The extra "and" will be removed.

6. L248: I think it is better to use irrigation depth or height instead of "irrigation doses"

The sentence will be modified as suggested by the reviewer.

**References:**

Dari, J., Brocca, L., Quintana-Seguí, P., Escorihuela, M.J., Stefan, V., and Morbidelli, R.: Exploiting high-resolution remote sensing soil moisture to estimate irrigation water amounts over a Mediterranean region, Remote Sens., 12, 2593, https://doi.org/10.3390/rs12162593, 2020.

Dari, J., Quintana-Seguí, P., Morbidelli, R., Saltalippi, C., Flammini, A., Giugliarelli, E., Escorihuela, M.J., Stefan, V., and Brocca, L.: Irrigation estimates from space: Implementation of different approaches to model the evapotranspiration contribution within a soil-moisture-based inversion algorithm, Agric. Water Manag., 265, 107537, https://doi.org/10.1016/j.agwat.2022.107537, 2022.

Jalilvand, E., Tajrishy, M., Hashemi, S.A.G., and Brocca, L.: Quantification of irrigation water using remote sensing of soil moisture in a semi-arid region, Remote Sens. Environ., 231, 111226, https://doi.org/10.1016/j.rse.2019.111226, 2019.

Modanesi, S., Dari, J., Massari, C., Tarpanelli, A., Barbetta, S., De Lannoy, G., Gruber, A., Lievens, H., Bechtold, M., Quast, R., Vreugdenhil, M., Zribi, M., Le Page, M., and Brocca, L.: A comparison between satellite- and model-based approaches developed in the ESA Irrigation+ project framework to estimate irrigation quantities. 2021 IEEE International Workshop on Metrology for Agriculture and Forestry (MetroAgriFor), 268-272, doi: 10.1109/MetroAgriFor52389.2021.9628453, 2021.